# Artificial Intelligence in the Diagnosis and Treatment of Pancreatic Cystic Lesions and Adenocarcinoma

**DOI:** 10.3390/cancers15092410

**Published:** 2023-04-22

**Authors:** Joanna Jiang, Wei-Lun Chao, Stacey Culp, Somashekar G. Krishna

**Affiliations:** 1Department of Internal Medicine, Ohio State University Wexner Medical Center, Columbus, OH 43210, USA; 2Department of Computer Science and Engineering, The Ohio State University, Columbus, OH 43210, USA; 3Department of Biomedical Informatics, The Ohio State University College of Medicine, Columbus, OH 43210, USA; 4Division of Gastroenterology, Hepatology, and Nutrition, Department of Internal Medicine, Ohio State University Wexner Medical Ceter, Columbus, OH 43210, USA

**Keywords:** artificial intelligence, pancreatic ductal adenocarcinoma, pancreatic cysts, endoscopy, IPMN

## Abstract

**Simple Summary:**

Pancreatic cancer will soon become the second leading cause of cancer-related death mainly due to a lack of early diagnosis. Artificial intelligence is being applied in various aspects of diagnosing medical conditions. In this review, we summarize the current literature on the application of artificial intelligence in the diagnosis and management of premalignant lesions that would otherwise progress to pancreatic cancer.

**Abstract:**

Pancreatic cancer is projected to become the second leading cause of cancer-related mortality in the United States by 2030. This is in part due to the paucity of reliable screening and diagnostic options for early detection. Amongst known pre-malignant pancreatic lesions, pancreatic intraepithelial neoplasia (PanIN) and intraductal papillary mucinous neoplasms (IPMNs) are the most prevalent. The current standard of care for the diagnosis and classification of pancreatic cystic lesions (PCLs) involves cross-sectional imaging studies and endoscopic ultrasound (EUS) and, when indicated, EUS-guided fine needle aspiration and cyst fluid analysis. However, this is suboptimal for the identification and risk stratification of PCLs, with accuracy of only 65–75% for detecting mucinous PCLs. Artificial intelligence (AI) is a promising tool that has been applied to improve accuracy in screening for solid tumors, including breast, lung, cervical, and colon cancer. More recently, it has shown promise in diagnosing pancreatic cancer by identifying high-risk populations, risk-stratifying premalignant lesions, and predicting the progression of IPMNs to adenocarcinoma. This review summarizes the available literature on artificial intelligence in the screening and prognostication of precancerous lesions in the pancreas, and streamlining the diagnosis of pancreatic cancer.

## 1. Introduction

While survival rates for lung, breast, and colorectal cancers (the three leading cancers in the US) have steadily improved over the past two decades, pancreatic cancer continues to carry a dismal prognosis, with 5-year survival rates of only 10% [1,2]. Due to the absence of symptoms in early disease, most pancreatic cancers (80–85%) are diagnosed after they have already metastasized or become unresectable [3].

The main opportunity for improving pancreatic cancer survival lies in its early diagnosis and surgical resection, highlighted by the comparatively high 5-year overall survival rates for those with stage 0 and 1A PDAC (85.8% and 68.7%, respectively) [4]. The most common pancreatic cancer is pancreatic ductal adenocarcinoma (PDAC). The pathways to PDAC are illustrated in Figure 1. PDACs typically first present as pre-malignant lesions. The most common precursor lesion is pancreatic intraepithelial neoplasia (PanIN), which accounts for 75–80% of PDACs. The progression of PanIN to PDAC has been well characterized; however, this is a pathologic diagnosis and not detectable in current imaging modalities [5]. Due to diagnostic technology limitations, PanINs do not represent viable options for pancreatic cancer screening.

Other PDACs (15–20%) originate as cystic lesions, most commonly intraductal papillary mucinous neoplasms (IPMNs), which account for half of all pancreatic cystic lesions (PCLs) [6]. IPMNs are mucin-producing epithelial tumors with papillary architecture on histology, and may progress to high-grade dysplasia and pancreatic cancer [7]. They are classified by location. The most common type is branch-duct (BD)-IPMN, a cyst that communicates with the main pancreatic duct [6]. Main-duct (MD)-IPMNs represent the main pancreatic duct dilation without other causes of obstruction. Mixed IPMNs display features of both types. Mucinous cystic neoplasms (MCNs), another type of mucinous cyst, also carry a risk of malignancy.

Non-mucinous pancreatic cysts include solid pseudopapillary neoplasms (SPNs) and cystic neuroendocrine tumors, representing less aggressive tumors than PDAC. Benign non-mucinous cysts include serous cystadenomas and lymphoepithelial cysts. The risk of malignant transformation in serous cystadenomas is less than 0.1% [8].

### 1.1. Challenges in Predicting Progression to PDAC

It is not recommended for the general population to undergo screening for PDAC due to the low overall prevalence and high risk of overdiagnosis and unnecessary intervention. Premalignant lesions are typically discovered incidentally on abdominal imaging for other clinical indications [6]. Up to 13% of all cross-sectional abdominal imaging studies in asymptomatic subjects will identify a pancreatic cyst, with higher rates for older patients. This creates the diagnostic challenge of differentiating benign cysts from those with malignant potential [9].

Pre-operative histopathologic detection of advanced neoplasia in IPMNs is challenging [10]. Endoscopic ultrasound-guided fine-needle aspiration (EUS-FNA) of PCLs and cyst fluid analysis are standard-of-care diagnostic modalities. Multiple case series highlight the inadequacy of these techniques in accurately identifying malignant lesions. A 2014 meta-analysis determined that although EUS FNA-based cytology had a high specificity of 90.6% (95% confidence interval 0.81–0.96), it had an intolerably low sensitivity of 68.4% (0.44–0.82) for distinguishing malignant from benign IPMNs [11].

While the goal of surgery is to resect lesions with advanced neoplasia (high-grade dysplasia and/or adenocarcinoma), approximately 42–63% of resected IPMNs were found to have only low-grade dysplasia [12,13,14,15,16,17]. Conversely, a negative biopsy or cytology does not rule out the presence of a high-risk lesion, and it is estimated that 5% of patients with IPMNs have concomitant adenocarcinoma elsewhere in the pancreas [18,19]. Considering the high morbidity of resection techniques (including Whipple procedures, left pancreatectomy, or total pancreatectomy) [14], there is a dire need for a reliable, accurate, and minimally invasive diagnostic tool for risk stratification of precancerous lesions.

### 1.2. Artificial Intelligence

With the abundance of clinical, radiographic, genomic, and endoscopic information becoming more readily available, there is a unique role for artificial intelligence (AI) in integrating and leveraging these data to diagnose pancreatic cancer and risk stratify lesions. AI is a mathematical application that automates pattern recognition and learning. It can transform large amounts of data from a reference set into clinically actionable conclusions.

AI techniques range in complexity from machine learning (ML) algorithms to deep learning (DL). While simple models have been used since the 1960s in the form of logistic regression, recent decades have seen the development of sophisticated neural network algorithms to predict risk for breast, lung, and other cancers [20,21,22,23,24,25,26,27]. The area under the curve (AUC) is a commonly used metric in risk modeling. AUC ranges from 0.50 (random chance) to 1.0 (perfect prediction).

## 2. Methods

Primary literature describing novel artificial intelligence algorithms to predict outcomes relating to PDAC or precancerous pancreatic lesions was evaluated. A literature search was conducted via PubMed and Embase using the terms “artificial intelligence”, “machine learning”, “radiomics”, or “deep learning” combined with “PDAC”, “pancreatic cystic lesions”, or “endoscopic ultrasound”. Articles were included if they described AI or ML algorithms that predicted clinically significant endpoints such as diagnosis of various PCL subtypes, diagnosis of PDAC, response to treatment, or overall survival.

We included studies published within the past ten years (2013–2023) that reported measures of accuracy such as AUC, accuracy, sensitivity, or specificity. Earlier articles utilized simple machine learning algorithms such as logistic regression; however, there was a drastic shift toward deep learning and neural networks after 2020. Case reports, protocols, meta-analyses, and review articles were excluded from our analysis.

## 3. Results

### 3.1. Developing a Screening Strategy

Risk factors for pancreatic cancer include cigarette smoking, obesity, chronic pancreatitis, advanced age, family history, and hereditary cancer syndromes [28,29,30]. Other patient characteristics such as diet, medication use, and infections have been investigated as possible contributors to PDAC development [31,32,33]. Despite extensive research into novel predictors and effect sizes of known risk factors, there is currently no unified recommendation on a high-risk population (outside of those with hereditary cancer syndromes) that may benefit from dedicated screening [34]. However, AI models have shown promise in incorporating multiple demographic and clinical characteristics to identify such a population.

#### 3.1.1. Models Incorporating Clinical Data

In a European study, Malhotra et al. (2021) used primary care data to train a logistic regression model to predict pancreatic cancer development in a cohort of patients aged 15–99 years [35]. The algorithm used documented symptoms and medical history to identify patients at a “high risk” of developing pancreatic cancer. Parameters included cardiovascular comorbidities, gastrointestinal disorders, and symptoms. The AUC was 65.6% for those under 60 years and 60.9% for older patients. The authors estimated that such an algorithm could result in earlier diagnosis for around 60% of pancreatic tumors. These models could be utilized for identifying an at-risk population.

New-onset diabetes has been suggested as a risk factor or potential early predictor of PDAC in multiple studies [36,37,38,39]. The pathophysiology of this link is complex; longstanding diabetes is a known risk factor for PDAC [30], and new evidence suggests that pancreatic cancer may also cause diabetes through a paraneoplastic syndrome or direct effects on islets and insulin secretion [40]. Using a discovery cohort of patients with new-onset diabetes, Sharma et al. (2018) created the END-PAC model to predict the development of pancreatic cancer within 3 years of diabetes onset [41]. The AUC of this logistic regression model was 0.87, and sensitivity and specificity were 80%. Significant predictors included weight change, change in blood glucose, and age of onset of diabetes. A newly published 2023 study by Chen et al. used random forest models to predict risk in patients with elevated glycated hemoglobin (HbA1c) [42]. When age, weight change, and other clinical variables were included, the sensitivity, specificity, and positive predictive value for patients in the top 20% of predicted risk were 60%, 80%, and 2.6%, respectively.

#### 3.1.2. Models Incorporating Genomics and Radiomics

Genetic mutations in the adenoma–carcinoma sequence leading to PDAC are well documented. KRAS and GNAS mutations are known to drive the progression of IPMNs [43]. When combined with the inactivation of tumor suppressor genes such as TP53 and SMAD4, these mutations cause aggressive tumor growth. Certain single-nucleotide polymorphisms have also been suggested as being linked to PDAC [44,45].

Klein et al. developed a logistic regression model incorporating genetic data to predict pancreatic cancer risk in a non-Hispanic population with European ancestry [46]. Patient and family history were included. The AUC was 58% for predicting cancer development, which increased to 61% when genetic factors (single-nucleotide polymorphism data) were included in the model. Building upon these findings, a 2022 Japanese study proposed a strategy to identify patients who may benefit from screening for early disease [9]. Their logistic regression model was designed using data from 35 patients with early-stage PDAC, and incorporated clinical indicators (tumor biomarkers, elevated pancreatic enzymes, pancreatitis history), risk factors (family history, diabetes, smoking history), and imaging findings (main pancreatic duct dilation). This model achieved an AUC of 0.67 for detecting early-stage PDAC in patients for whom imaging was “strongly recommended” based on the proposed screening strategy. When such patients had notable imaging findings, the AUC rose to 0.80.

While CT and MRI classically provide qualitative data that are interpreted by radiologists, images can also be considered as a matrix with large amounts of quantitative data. Qureshi et al. published a novel study in identifying high-risk individuals based on “pre-diagnostic” CT imaging, acquired 6 months to 3 years before patients were diagnosed with PDAC. These imaging studies were performed prior to the development of qualitative signs of cancer that could be detected by trained radiologists [47]. Textural and morphological features of the pancreas were analyzed from a set of 66 contrast-enhanced abdominal CT scans. The naïve Bayes classifier (ML algorithm) was able to classify scans into pre-diagnostic vs. control (no PDAC diagnosis) groups with 86% accuracy in an external dataset. The generalizability of this model, however, was limited by its relatively small dataset.

In summary, existing AI models appear to be of limited immediate clinical utility in dictating screening guidelines for the general population, with the AUC hovering around 0.60. Their reliability increases in populations already determined to be at higher risk, such as those with new-onset diabetes, and they do show promise in identifying subsets of these patients who may benefit from screening. Predictive ability increases with the addition of genomics and radiomics into ML models.

### 3.2. Detection and Risk Stratification of Pancreatic Cystic Lesions

Professional societies provide diverging recommendations for diagnosing and monitoring incidentally discovered PCLs. There is variation in the timing of cross-sectional imaging for diagnosis and surveillance, indications for EUS and EUS-guided (fine-needle aspiration), the role of cyst fluid biomarkers and other advanced technology, indications for surgery, and surveillance intervals. The Fukuoka/International Consensus guidelines provide a risk stratification structure for IPMNs, stratifying them as “high-risk stigmata” and “worrisome features”11]. EUS and EUS with FNA are recommended by the Fukuoka guidelines in the presence of worrisome features. Both the American College of Gastroenterology (ACG) and Fukuoka guidelines recommend EUS if there is preceding acute pancreatitis [11,48]. The American Gastroenterology Association (AGA) recommends EUS-FNA for cysts with at least two high-risk features (size ≥ 3 cm, dilated main pancreatic duct, or intracystic solid component) [48]. Conflicting recommendations between the three societies remain regarding screening intervals and characteristics of “high-risk” lesions.

#### 3.2.1. Cross-Sectional Imaging in IPMNs

One reason for differing guidelines is the availability of vast amounts of clinical, radiographic, and endosonographic findings of varying clinical significance that could be taken into account. Table 1 summarizes studies utilizing AI models for the risk stratification of IPMNs. While the rates of malignancy in IPMNs is well characterized (ranging from 57 to 92% for MD-IPMN and 6 to 46% for BD-IPMNs), few reliable tools exist for predicting the risk of an individual lesion progressing to carcinoma [49]. Multiple nomograms have been proposed that incorporate factors ranging from biomarkers and radiologic features to immunofluorescent images of resected cysts [50,51,52].

Machine learning algorithms have demonstrated the ability to integrate this multitude of information to more reliably risk-stratify PCLs, particularly IPMNs. In a 2016 proof-of-concept study, Permuth et al. demonstrated the potential of texture analysis of CT images to identify malignant IPMNs with an AUC of 0.77 [53]. This improved to 0.92 with the inclusion of microRNA genomic data (sensitivity 83%, specificity 89%). In comparison, “worrisome features”, as defined by consensus guidelines (main duct dilation, cyst size > 3 cm, cyst wall thickening, mural nodules, or acute pancreatitis), yielded an AUC of only 0.54. Hanania et al. reported similarly promising findings, with a logistic regression model achieving an AUC of 0.96 (sensitivity 97% and specificity 88%). These models had false positive rates of 5%, while the Fukouka criteria had a 36% false positive rate in their cohort [54]. Both studies were limited by small patient populations; only 38 patients (20 benign and 18 malignant IPMNs) were included in the first study, and 53 (19 benign and 34 malignant IPMNs) in the second.

In the past five years, more sophisticated AI algorithms have been leveraged for risk prediction. A retrospective study by Chakraborty et al. (2018) used CT scan features from 103 patients to predict IPMN dysplasia [55]. All patients underwent resection of BD-IPMNs, which were confirmed on pathology. Random forest and support vector machine (SVM) models were developed to categorize cysts into low or high risk (determined by grade of dysplasia at resection). Using radiographic features along with five clinical variables (age, gender, cyst size, presence of solid component, and symptoms), the random forest model yielded an AUC of 0.77 with a sensitivity and specificity of 0.68 and 0.84, respectively.

Corral et al. (2019) used convolutional neural networks (CNNs) to identify normal pancreas, low-grade dysplasia, and high-grade dysplasia/adenocarcinoma based on MRI radiomics features of IPMNs prior to pancreatectomy. The deep learning protocol achieved a sensitivity and specificity of 75% and 78%, respectively, with an AUC of 0.90 (95% confidence interval 0.71–0.85) for detecting high-grade dysplasia or cancer. In comparison, the AUC was 0.76 (0.70–0.84) for the AGA and 0.77 (0.70–0.85) for the Fukuoka guidelines in their cohort [56]. Interestingly, the authors found the Fukuoka criteria to be more sensitive and AGA to be more specific for detecting high-risk lesions.

#### 3.2.2. Cross-Sectional Imaging in PCLs

In a larger retrospective study including 214 patients who underwent resection for pancreatic cysts at Johns Hopkins, the radiomics-based random forest model yielded an AUC of 0.940 for distinguishing between five types of cystic neoplasms (IPMNs, MCNs, SPNs, SCAs, and cystic NETs) [57]. The radiomics model was compared to radiologists’ diagnostic interpretation; the AUC for academic radiologists reached 0.895. Predictions were made based on radiomics features from preoperative CTs and demographics (age and gender). Liang et al. reported similar success for their SVM and logistic regression models in differentiating between IPMNs, MCNs, and SCAs based on data from CT images. An SVM algorithm was used to train a fused radiomics–DL model, which yielded an AUC of 0.92 for the diagnosis of SCA and 0.97 for differentiating between MCNs and IPMNs [58].

Artificial intelligence has also been used to optimize images and improve the diagnostic accuracy of radiologist interpretation and reading. Matsuyama et al. (2022) applied deep learning reconstruction (DLR) to improve the quality of MRCP images obtained from 32 patients with IPMNs. The reference standard was determined through EUS, ERCP, or surgical histopathology. DLR was able to significantly improve the signal-to-noise ratio and contrast-to-noise ratio. One model improved reader accuracy from 70% to 78% for classifying IPMN subtypes [59]. Inter-observer agreement was almost perfect after image reconstruction, with a weighted Kappa statistic between 0.97 and 0.98 (*p* < 0.0001). Yamashita et al. (2021) took a unique approach to AI-guided diagnosis by using natural language processing (NLP) to identify patients with PCLs and extract lesion measurements from CT and MRI radiologist reports [60]. The true positive rate was 98.2% with a false positive rate of 3.0% when the model was compared against the consensus of two radiologists’ annotations.

#### 3.2.3. EUS-Guided Diagnostics

The morphology of PCLs observed during EUS has been utilized in AI models (Table 2). Vilas-Boas et al. (2022) developed a CNN algorithm to differentiate mucinous cysts (IPMNs and MCNs) from non-mucinous cysts (SCAs and pseudocysts) [61]. Cyst type was determined based on resected specimens, biopsy samples, or cyst fluid studies and cytology. Mucinous cysts were defined as having CEA fluid levels > 192 ng/mL and glucose < 50 mg/dL, or mucinous epithelial cells on cytology. EUS images from 28 pancreatic cysts were used to train the model, which achieved an AUC of 1 for identifying mucinous cysts. The overall accuracy was 98.5%, sensitivity 98.3%, and specificity 98.9%.

Similarly, EUS-delineated cyst morphology has also been applied to develop AI models for differentiating dysplasia in IPMNs [23,62,63]. Schultz et al. (2022) developed CNNs to distinguish between IPMNs with low-grade dysplasia and high-grade dysplasia/carcinoma (advanced neoplasia) [23]. A training set of EUS images was collected from 43 patients who underwent pancreatectomy for histologically proven IPMN. Surgical histopathology served as the reference standard. EUS images containing the lesion of interest were selected by taking every third or sixth video frame. The manual pre-processing of images included resizing and normalizing pixel values. The algorithm was able to classify advanced neoplasia with an impressive accuracy of 99.6%; this was higher than individual guidelines (AGA, ACG, and revised Fukuoka and European guidelines), which had accuracies between 51.8% and 70.3% for this cohort.

The last two decades have seen the development of novel biopsy and visualization using advanced endoscopy. Confocal laser endomicroscopy (CLE) is a technique where a mini-probe is advanced through an EUS-FNA needle [64]. Tissue is illuminated with a low-power laser, and a real-time, gray-scale image of the cyst epithelium at the microscopic level is created [65,66,67]. Endoscopists are able to obtain real-time optical biopsies of PCLs in multiple locations. Over the last decade, major trials (INSPECT, DETECT, CONTACT, INDEX) have established reference standards and safety profiles for EUS-nCLE in subjects with PCLs [68,69,70,71,72].

EUS-guided needle-based CLE (nCLE) can accurately diagnose IPMNs and identify high-grade dysplasia, and has the potential to risk-stratify cystic lesions [67,68,69,72,73,74]. This offers a significant benefit over classical pancreatic cyst sampling methods, which can be unreliable and carry a risk of pancreatitis and bleeding [75]. However, EUS-nCLE video files are typically large with multiple frames, and there is often interobserver variation. AI can offer the ability to interpret such large amounts of imaging data and detect advanced neoplasia.

Our team previously designed a CLE-based convolutional neural network (CNN)-artificial intelligence (AI) algorithm to risk-stratify BD-IPMNs (predict advanced neoplasia vs. low-grade dysplasia) [76]. EUS-nCLE video frames from 35 patients with histologically proven IPMNs were used to design two CNN models to measure papillary epithelial thickness (indicative of advanced neoplasia) and to extract nCLE features for risk stratification. The AI models had higher accuracy (82%) for the detection of HGD-Ca than the AGA guidelines (68.6%) and Fukuoka criteria (74.3%).

AI has been applied to histopathologic diagnosis after resection as well. Kriegsmann et al. developed three CNN algorithms for identifying and quantifying tissue categories such as PanIN and PDAC in whole-tissue slides from 201 patients who underwent surgical resection of pancreatic lesions [77]. The models achieved a balanced accuracy of 92.1%.

**Table 2 cancers-15-02410-t002:** EUS-based models for evaluating PCLs.

Study	Sample Size	Model	Task	Accuracy	Comparisons
Schultz, 2022 [23]	43	CNN	Low- vs. high-grade IPMN dysplasia	Accuracy 99.6%	Higher accuracy than AGA, ACG, Fukuoka guidelines
Kuwahara, 2019 [62]	50	Deep learning	Evaluate malignant potential in IPMN images	Sensitivity 95.7%, Specificity 96.2%, Accuracy 94.0%	Human interpretation, 56% accuracy
Nguon, 2021 [63]	109	CNN	Differentiate between MCNs and SCAs	Accuracy 83%	
Machicado, 2021 [76]	35	CNN	Low- vs. high-grade BD-IPMN dysplasia	Accuracy 82%	Higher accuracy than AGA and Fukuoka guidelines

#### 3.2.4. Limitations and Future Directions

There are some limitations to imaging-based AI models that must be taken into account. First, because histopathology of resected lesions acts as the reference standard, most included patients would have required surgery, and AI models may be biased toward higher-risk populations. Second, most studies describe single-center algorithms that were not externally validated. Generalizability and reproducibility should be demonstrated with further multi-center analyses. Additionally, studies on AI-based risk stratification of PCLs have all been retrospective. Prospective validation studies are necessary to demonstrate their utility in clinical practice.

Multiple studies are also limited by small sample sizes; this may be due to the low overall number of patients who undergo surgical resection of PCLs within the study periods, as well as a lack of widespread adoption of EUS and nCLE (with the need for training on advanced endoscopy and novel nCLE imaging techniques).

AI offers useful guidance for clinical decision-making surrounding PCLs. An AI-based decision support system (DSS) is a computer program that utilizes artificial intelligence techniques to assist decision-makers in solving complex problems. It combines machine learning algorithms, statistical models, and decision analysis tools to provide recommendations to the user. The system may also provide simulations of different scenarios, predictive modeling, and visualization tools to help users understand the possible outcomes of their decisions. AI-based DSS has been utilized for predicting mental health disorders, recommending surgery, and diagnosing COVID-19, among other tasks [78,79,80,81,82,83,84]. With increasing clinical and imaging data on PCL subtypes collected, future studies in this field may leverage DSS as well.

#### 3.2.5. Radiomics in Detection of PDAC

AI has been applied to imaging analysis of lung, prostate, and breast cancer; however, the task of diagnosing pancreatic cancer remains particularly challenging [78,85,86,87,88]. The pancreas is highly variable in size, shape, and location. It lies in close proximity to organs of varying radiographic textures (including the liver, stomach, intestines, and spleen), and occupies comparatively little space in cross-sectional images. Pancreatic tumors often have similar characteristics as their background tissue, which impacts diagnostic efficiency [89]. Therefore, while AI algorithms for certain solid tumors are able to analyze minimally processed images, researchers often need to manually outline or segment the pancreas (divide it into four segments) prior to applying AI techniques [47,55]. This increases specialist workload and time to arrive at a diagnosis. A group at Zhejiang University was able to create a deep learning model that automated image processing and analysis [90]. Using nearly 150,000 abdominal CT images from 319 patients, the model was able to diagnose pancreatic tumors and propose treatment with an accuracy of 82.7% for all pancreatic tumor types. The model yielded an AUC of 0.87. Notably, there was higher accuracy for identifying PDAC (87.6%), and perfect accuracy (100%) for IPMNs.

Chen et al. developed an SVM prediction algorithm to predict PDAC in chronic pancreatitis patients based on CT images taken prior to cancer diagnosis [91]. The algorithm outperformed expert image review, achieving 100% accuracy and an AUC of 1.00 in chronic pancreatitis patients. For patients without pancreatitis, the accuracy was still high at 94–95% with an AUC of 0.98–0.99.

### 3.3. AI in Pancreatic Cancer Prognostication

#### 3.3.1. Treatment Selection

For patients with resectable disease, standard-of-care therapy involves surgery and adjuvant chemotherapy (typically FOLFIRINOX, which comprises fluorouracil, irinotecan, leucovorin, and oxaliplatin) [92,93]. Locally advanced and unresectable cancers can be treated through systemic chemotherapy and radiation. With the identification of common mutations driving PDAC and an increased understanding of the tumor microenvironment, there has been increased interest in immune checkpoint inhibitors and immunotherapy as novel treatment options [94].

Proof-of-concept studies have provided promising evidence for AI in predicting responses to various treatments. Liang et al. (2021) used a multivariate Cox regression model to identify radiomics features from pre-operative MRIs that could predict individual response to a new oral therapy (S-1) as an adjuvant chemotherapy and postoperative disease-free survival [95]. Tumor location and another imaging feature were significant predictors of S-1 efficacy, with mass in the pancreatic head being associated with lower treatment efficacy and survival.

Iwatate et al. (2020) developed random forest models with XGBoost to predict abnormal p53 and PD-L1 expression, as well as survival, based on imaging features from CT scans [96]. P53 mutations are associated with more aggressive PDAC [97], and PD-L1 is a target for immunotherapies. Imaging features from CT scans of 107 patients with surgically resected PDAC were used to train the AI models, which achieved an AUC of 0.795 and 0.683 for predicting the p53 and PD-L1 status, respectively. The authors also confirmed that these mutations predicted lower overall survival in their population.

#### 3.3.2. Survival Prediction

Although 20% of PDAC patients present with “resectable” lesions at diagnosis, approximately a third of patients who undergo surgery develop recurrence after resection [98,99,100]. The American Joint Committee on Cancer (AJCC) uses the tumor, node, metastasis (TNM) staging system to classify PDAC, with each stage carrying a different prognosis [101,102]. Various AI models have been developed to predict prognosis in other solid tumors [103,104]. Despite the multitude of clinical and radiographic data collected during routine care and cancer treatment, there is currently no individualized, reliable risk score to assess survival and recurrence.

Lee et al. (2022) developed a random forest model to predict overall survival and recurrence-free survival for patients who underwent PDAC resection [105]. Using preoperative clinical data and CT images, the model yielded an AUC of 0.76 for 2-year overall survival and 0.74 for 1-year recurrence-free survival. This was comparable to established guidelines from the AJCC. Another random forest model by Kaissis et al. reported a higher AUC of 90% with 87% sensitivity and 80% specificity for the prediction of above- versus below-median overall survival for a similar population of PDAC patients. The model used radiomics data from MRIs [106].

EUS-FNB data have been used to predict PDAC prognosis more accurately. Park et al. (2021) obtained frequencies of gene mutations through targeted sequencing of EUS-FNB specimens from PDAC patients [107]. Their Cox regression model successfully classified patients into a high-risk group (median overall survival 5.95 months) and a low-risk group (median survival 15.27 months) based on genetic mutations detected from cyst fluid and clinical factors such as age, stage, and mass size. The hazard ratio was 6.06 (*p* < 0.001).

## 4. Discussion

Multiple retrospective studies have highlighted the potential of machine learning in advancing pancreatic cancer screening and prognostication and improving the diagnosis and risk stratification of PCLs, particularly IPMNs. Some AI algorithms have shown promise in identifying high-risk patients who may benefit from PDAC screening, while others were able to accurately diagnose and risk-stratify pancreatic cysts, anticipate response to treatment, and predict patient survival (summarized in Figure 2).

Earlier studies were centered around logistic regression models to identify predictive factors for PDAC, while more complex decision trees and neural networks have demonstrated greater accuracy in some recent studies. The primary benefit of AI lies in the ability to predict outcomes based on vast amounts of radiographic, clinical, and endoscopic information. Such data can be obtained in an inexpensive and minimally invasive manner as part of routine care, potentially reducing the need for high-morbidity surgical resections.

Although recent studies continue to provide evidence against population-based screening for PDAC, AI can help identify subsets of patients at higher risk. Machine learning application to CT or MRI images has proven to be a helpful adjunct to radiologist reads. Its predictive ability has sometimes surpassed that of trained specialists or international guidelines. AI algorithms applied to advanced techniques such as EUS-nCLE have demonstrated greater accuracy than the current SOC in diagnosing and risk-stratifying IPMNs. After the diagnosis of PDAC, AI may play a role in identifying a personalized treatment plan, as well as predicting prognosis.

### Future Directions

Current challenges to creating accurate, scalable AI models for PDAC include insufficient data for model training, the subsequent risk of overfitting with prediction models, and the need for specialized, resource-rich hospitals with large patient populations to conduct such studies. There remains a need for multi-center studies that include diverse cohorts to improve the generalizability of these algorithms. Additionally, the majority of radiomics algorithms in this review required manual preprocessing of images, which can be time-consuming for specialists. More sophisticated models that can be applied to unedited images or videos would reduce long-term healthcare utilization.

Existing algorithms that risk-stratify PCLs may also be biased toward higher-risk lesions; due to the absence of a reference standard other than surgical histopathology, only patients who eventually required surgical resection were included in these studies. Radiomics-based models may, therefore, overestimate risk when applied to a newly discovered, undifferentiated lesion. The large proportion of patients in these studies who were found to have benign PCLs also highlights the inadequacy of standard tools in predicting malignant potential. Future models need to integrate all SOC variables, cyst fluid analysis, and nCLE imaging to identify effective combinations of available data for accurate diagnosis.

## 5. Conclusions

AI algorithms have been developed to identify high-risk populations who may benefit from PDAC screening, determine malignant potential of PCLs, and predict treatment response and cancer survival. Models for PCL risk stratification have demonstrated high accuracies, while algorithms for predicting an individual’s risk of developing PDAC were less reliable. There remains a role for more sophisticated algorithms that require minimal data pre-processing, as well as models developed using diverse, multi-center cohorts.

## Figures and Tables

**Figure 1 cancers-15-02410-f001:**
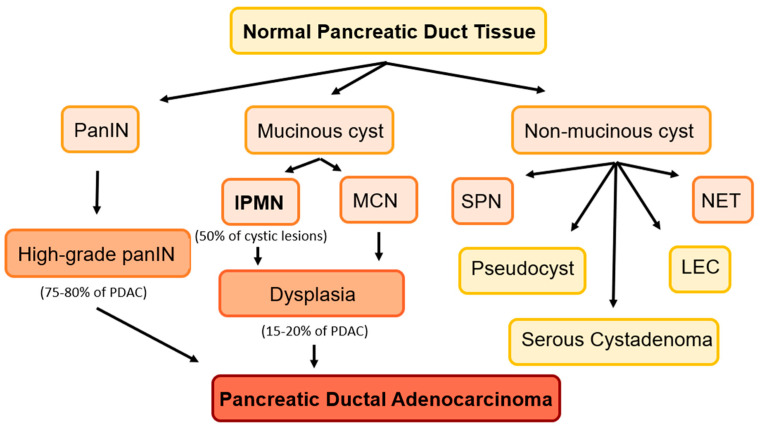
Precancerous lesions of the pancreas. Mucinous cysts: IPMNs and mucinous neoplasms (MCN). Non-mucinous cysts: serous pseudopapillary neoplasms (SPN), lymphoepithelial cysts (LEC), and cystic neuroendocrine tumors (NET). Pseudocysts, serous cystadenomas, and LEC are benign lesions.

**Figure 2 cancers-15-02410-f002:**
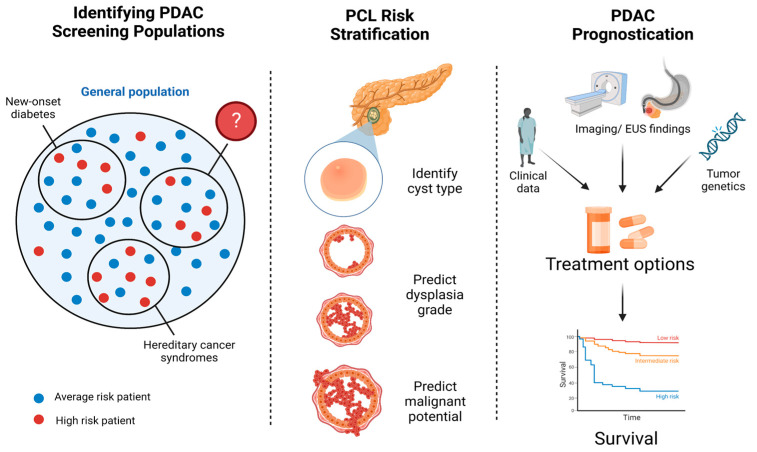
AI developments in PDAC diagnosis and treatment. Machine learning has been leveraged to identify high-risk populations for PDAC screening, risk-stratify incidentally discovered PCLs, and assist with PDAC prognostication.

**Table 1 cancers-15-02410-t001:** Radiomic models for diagnosis and risk stratification of IPMNs.

Study	Sample Size	Data	Best-Performing Model	Task	AUC	Comparisons
Permuth, 2016 [53]	38	CT texture analysis + genomics	Logistic regression	Distinguish malignant from benign IPMNs	0.92	N/A
Hanania, 2016 [54]	53	CT imaging (texture, shape, intensity)	Logistic regression	IPMN high- vs. low-grade dysplasia	0.96	Lower false positive rate than Fukuoka
Chakraborty, 2018 [55]	103	CT imaging features	Random forest	High- vs. low-risk BD-IPMN	0.77	N/A
Corral, 2019 [56]	139	MRI imaging features	CNN	Identify high-grade dysplasia or cancer in IPMNs	0.78	Accuracy was comparable to AGA/Fukuoka
Chu, 2022 [57]	214	CT radiomics features	Random forest	Classify mucinous and non-mucinous cysts	0.94	Accuracy was comparable to radiologist
Liang, 2022 [58]	193	CT + clinical data	Fused radiomics-DL	Differentiate MCN from IPMN	0.973	N/A

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
