# Peer review of "Artificial Intelligence in the Diagnosis and Treatment of Pancreatic Cystic Lesions and Adenocarcinoma"

_cancers, 2023, doi:10.3390/cancers15092410_

Round 1

Reviewer 1 Report

The paper reviewed AI based methods and application in the diagnosis and treatment of pancreatic cystic lesions (PCLs) and adenocarcinoma. The review is well organized and clearly depicts how AI methods and machine learning models are leveraged in predicting progression, identifying high-risk groups for PCLS, guided diagnosis, and treatment recommendation by incorporating clinical data, genomic and radiomics. However, I have several major concerns on the review. 

1. The literature selection strategy and criteria is not clearly defined.

2. The authors majorly reviewed traditional machine learning models applied in diagnosing and treating PCLs. I am wondering the review covered sufficient works and reference in the domain. It lacks summary on AI based decision support systems, knowledge representations and Natural Language Application and deep learning methods being applied in the domain.

3. The limitation and potential chance of AI based methods were not well presented and discussed in PCLs domain. 

Author Response

Thank you for your time and helpful suggestions. We have addressed the comments as follows:

Point 1. The literature selection strategy and criteria is not clearly defined.
Response: We expanded our methods section to describe our inclusion and exclusion criteria. We describe the specific article types and clinical outcomes included in our initial literature search. (lines 101-109)

Point 2. The authors majorly reviewed traditional machine learning models applied in diagnosing and treating PCLs. I am wondering the review covered sufficient works and reference in the domain. It lacks summary on AI based decision support systems, knowledge representations and Natural Language Application and deep learning methods being applied in the domain.
Response: We appreciate the feedback regarding areas of AI which were not discussed in as much detail in our review. We were able to find a large number of studies using traditional ML models. To address this point, we expanded our search to include the terms “natural language,” “decision support systems,” and “knowledge representations.” We have added an additional reference [62] which discusses Natural Language Processing in the PCLs domain; this is described in lines 253-258. However, we were not able to find articles on PubMed or Embase which discuss more advanced AI models which serve as decision support systems or can perform complex tasks such as knowledge representation and reasoning. This presents interesting future directions in AI in pancreatic cancer research; we discuss the potential for AI-based DSS in this domain in lines 316-325.

Point 3. The limitation and potential chance of AI based methods were not well presented and discussed in PCLs domain.
Response: We have added a discussion of limitations and future directions at the end of our PCLs section, including possible biases in the included studies and concerns regarding generalizability and reproducibility (lines 302-313).

Reviewer 2 Report

General remarks

In this paper, the authors reviewed the published papers on artificial intelligence in the diagnosis and Treatment of Pan-2 creatic Cystic Lesions and Adenocarcinoma. This is an extensive review, and the paper may give an overview of this aspect.

Specific remarks

The authors should provide a table highlighting the neural network based on Endoscopic ultrasound-guided fine needle aspiration.

Similarly, a table should provide on the survival of the patients and the application of neural networks.

Author Response

Point 1. The authors should provide a table highlighting the neural network based on Endoscopic ultrasound-guided fine needle aspiration.
Response: We have added a table (line 299) summarizing studies which described deep learning/ AI models using EUS data. We describe their notable findings, as well as comparisons of the proposed models to current standards of care.

Point 2. Similarly, a table should provide on the survival of the patients and the application of neural networks.
Response: Although there were many studies on survival prediction in other solid tumors, we were unfortunately only able to find a small number of articles which describe AI models to predict survival specifically in PDAC patients. We discuss three articles in lines 369-383, but we were not able to make a table describing applications of neural networks in predicting patient survival given the small number of studies.

Round 2

Reviewer 1 Report

The review is an important and interesting summary for AI applications and methods in treatment of pancreatic cystic lesions and adenocarcinoma. It would be better if the time range, volume covered and AI topic trend being included in the articles selection section.

Author Response

Dear Reviewer,

Thank you for taking the time to provide feedback. To the suggestion "It would be better if the time range, volume covered and AI topic trend being included in the articles selection section," we have expanded our Methods section (added lines 109-112) to include that we focused on articles published within the past 10 years, after 2013. We also comment that older articles utilized primarily logistic regression and there was a drastic shift toward more advanced AI after 2020.